# Natural History and Risk Factors of Hymenoptera Venom Allergy in Dogs [note 1]

**DOI:** 10.3390/ani14223220

**Published:** 2024-11-10

**Authors:** Edwin Chapman, Erin Ashley West, Mitja Kosnik, Nina Maria Fischer, Claude Favrot, Leo Beeler, Ana Rostaher

**Affiliations:** 1Dermatology Unit, Clinic for Small Animal Internal Medicine, Vetsuisse Faculty, University of Zurich, 8057 Zurich, Switzerland; edwin.chapman@uzh.ch (E.C.); nfischer@vetclinics.uzh.ch (N.M.F.); cfavrot@vetclinics.uzh.ch (C.F.); 2Clinical Trials Unit, Kantonal Hospital St. Gallen, 9007 St. Gallen, Switzerland; erin.west@kssg.ch; 3Division of Allergy, University Clinic of Respiratory and Allergic Diseases Golnik, 4204 Golnik, Slovenia; mitja.kosnik@klinika-golnik.si; 4Faculty of Medicine, University of Ljubljana, 1000 Ljubljana, Slovenia; 5Swiss Economic Institute, Federal Institute of Technology Zurich, 8092 Zurich, Switzerland; leoserafino.beeler@uzh.ch

**Keywords:** dog, anaphylaxis, natural history, venom immunotherapy

## Abstract

Hymenoptera venom allergy (HVA) is a potentially life-threatening systemic hypersensitivity reaction. In this study, data from 178 dogs with insect sting allergic reactions were analyzed and several risk factors for severe systemic reactions to Hymenoptera stings were identified. Furthermore, a significant number of dogs suffered subsequent systemic reactions to Hymenoptera stings, indicating that venom immunotherapy may be a valuable intervention to prevent future reactions. This study should raise the awareness of dog owners that Hymenoptera stings are associated with HVA and its possible consequences.

## 1. Introduction

Hymenoptera venom allergy (HVA) is a potentially life-threatening condition triggered by stings from honeybees, wasps, or ants. The injected allergens can potentially lead to local pain and swelling, larger regional reactions or severe systemic reactions (SSRs) due to an immediate Immunoglobulin E (IgE)-mediated hypersensitivity response [1]. Up to 94.5% of humans report having been stung at least once in their lifetime [2]. The prevalence of systemic reactions due to Hymenoptera stings differs according to the geographic location and the different allergen exposure of a population. While in the USA, 0.5% to 3.3% of adults experience a systemic reaction, European studies report a broader prevalence ranging from 0.3% to 7.5% [3,4]. Insect stings have been identified as the principal cause of anaphylaxis (59%) in some regions, followed by drugs (18%) and food (19%) [5]. In veterinary medicine, similar findings have been observed, with a study at the Vetsuisse Faculty of the University of Zurich reporting 41.1% of all anaphylactic reactions in dogs being caused by insect stings [6]. While in Germany, about 20 human deaths per year due to insect venom anaphylaxis are recorded, Switzerland has about twice as many deaths in relation to the population [7]. Currently, there are no data regarding the mortality rate in canine patients with severe HVA.

To date, the diagnosis of a severe reaction to Hymenoptera venom is achieved by a thorough anamnesis and observation of a typical clinical pattern [8]. Treatment of a systemic allergic reaction is based on the severity of the clinical signs. While mild reactions involving only cutaneous symptoms can be treated with antihistamines and glucocorticoids, the first-line treatment for an anaphylactic shock is epinephrine [9]. If a patient suffers a severe reaction, the detection of venom-specific IgE antibodies in the blood or skin test may be performed to discover or confirm the culprit allergen and build a basis on which venom immunotherapy (VIT) may be carried out [10]. VIT is the only treatment shown to reliably prevent further anaphylactic episodes and improve quality of life (QOL) compared to those relying solely on adrenaline auto-injectors [11,12,13]. Human studies report an efficacy of 80–95% and few side effects, with similar results published in dogs undergoing VIT [14,15,16,17,18,19,20].

Studies investigating the natural history of HVA in humans have shown that patients with a history of cutaneous reactions typically experience the same or a less severe response to subsequent stings. However, adults with a history of SSR after a Hymenoptera sting have a 40–60% likelihood of experiencing another systemic reaction [3,21,22]. To date, no comparable studies concerning the natural history of dogs with HVA have been carried out in veterinary allergology.

In humans, several key risk factors for SSRs have been identified including clonal mast cell diseases (CMD) and elevated baseline serum tryptase levels [23,24,25,26,27]. Older age is also a significant risk factor, with fatalities being rare before age 50 and patients over 40 being more susceptible to SSRs [27,28]. Furthermore, a well-tolerated sting within the previous two months prior to a subsequent insect sting increases the risk of a systemic reaction by nearly 60% [4]. The evidence is conflicted on other risk factors including male sex and the type of venom injected, with some studies suggesting that wasp stings provoke more severe reactions [14,25,27,29,30]. Over the past decade, previously suspected risk factors such as concurrent use of antihypertensive medications, cardiovascular disorders, pulmonary conditions, and stings to the head and neck regions have been found not to increase the severity of reactions [14,27,30,31]. A case series of dogs with cutaneous immediate-type hypersensitivity reactions found that those with a history of atopic dermatitis (AD) were at a greater risk of developing urticaria or anaphylaxis (11 of 24) [6]. Several studies have associated a smaller body weight in dogs with anaphylaxis, while robust data on sex, age, or breed predilections remain limited [6,32,33,34].

Despite the wealth of information on HVA in humans, there remains a lack of data regarding its occurrence in veterinary medicine, especially concerning risk factors for severe life-threatening cases as well as the natural history of these patients. Therefore, this study aims to fill this gap by analyzing HVA cases to gain insights into the disease incidence, natural history, and possible disease modifying risk factors. The secondary aims were to assess the clinical features and the impact of the disease on the life quality of the owners. We hypothesize that the prevalence and risk factors for systemic reactions in dogs mirror those observed in humans.

## 2. Materials and Methods

This retrospective epidemiological study analyzed dogs presented to the Department of the Clinic for Small Animal Internal Medicine and Emergency Clinic at Vetsuisse Faculty, University of Zurich, between January 2018 and December 2022. The database query included the following keywords: insect, bee, bee sting, wasp, wasp sting, insect bite, Hymenoptera.

### 2.1. Animal Selection and Definitions

Dogs were included if a reaction to Hymenoptera stings was diagnosed following admission. HVA diagnosis was based on the compatible history (observed or suspected Hymenoptera sting) and clinical signs consistent with IgE-mediated allergic reactions. Allergen-specific IgE testing was performed in some patients to confirm the suspected trigger. Additionally, before the final diagnosis was made, all possible differential diagnoses for urticaria and anaphylaxis were considered and excluded [6,35,36,37,38]). These are summarized in Table 1.

Disease severity was classified according to previously established grading systems, as outlined in Table 2 (Grades 1, 2, and 3 refer to mild, moderate, and severe reactions to Hymenoptera stings, respectively) [6,39].

### 2.2. Data Collection

Data were collected during two consecutive time periods, with immediate data collection encompassing signalment, history, clinical details (including hematology, biochemical profile, blood pressure, ultrasound, and radiographs) as well as sting-related information (time of day of the sting, body location, environment). All variables are displayed in Appendix A. To achieve higher statistical power, several variables were grouped: breed (purebred vs. mixed), duration of hospitalization (<6 h, 6–12 h, 12–48 h, >48 h), time of day (06–10:00, 10–14:00, 14–18:00, 18–06:00), number of stings (1, 2–3, >3), time between stings (<2 months, 2–12 months, 1–3 years, >3 years), and HVA severity (Grade 1, 2 and 3). Dog age at the time of Hymenoptera sting was calculated based on birth and presentation dates. Follow-up data included recurrent Hymenoptera stings, reaction severity, owner quality of life, adrenaline autoinjector use, and awareness of VIT. Follow-up lasted a minimum of one year to allow for enough time for potential subsequent stings to occur, with a shorter period if re-stings occurred earlier. Owners were contacted to evaluate the HVA natural history, assess their quality of life, provide guidance on preventative measures, and to test their ability to recognize the different types of Hymenoptera. The structure and composition of the questionnaire was modeled after validated human medical questionnaires [40,41]. The survey is displayed in Appendix A.

### 2.3. Statistical Analysis and Correlations

The incidence of Hymenoptera stings was calculated based on the total presentations at the Emergency Clinic and the Clinic for Small Animal Internal Medicine/Dermatology during the study period. The breed, sex, and age of the included population were compared to data from the Swiss dog registration system (ANIS) and institutional records.

Descriptive statistics (median, minimum, and maximum values) were used to report age, weight, number of episodes, and duration of hospitalization. The natural history of Hymenoptera stings was evaluated by the initial and recurrent sting episodes. The frequency of systemic reactions (grades 2–3) was calculated for both initial and repeat stings as well as the progression of dogs initially showing only skin symptoms (grade 1) to systemic reactions (grades 2–3). The natural history of HVA was assessed before the initiation of VIT.

To identify risk factors for a more severe reaction to Hymenoptera stings, bivariate analysis (Fisher’s exact test) and multivariable analysis (ordinal regression) were employed to calculate the adjusted odds ratio (OR) and 95% confidence interval (CI). The variables assessed in the bivariate analysis were: age under two years, weight under 10 kg, male sex, purebred dogs, more than one episode, sting interval under two months, a previous systemic reaction, time of day, sting in the oral cavity, bee venom, canine atopic dermatitis, and comorbidities. Variables with clinical relevance and sufficient data were further assessed in the multivariable analysis including age under two years, weight under 10 kg, male sex, purebred status, multiple episodes, sting in the oral cavity, bee venom, and comorbidities. The ordinal regression analyzed factors associated with reaction severity were categorized into three grades, 1 (mild), 2 (moderate), and 3 (severe), in order to examine which risk factors best predicted more severe reactions. All statistical analyses were conducted using R version 4.3.1, with a significance threshold set at a *p*-value of <0.05.

## 3. Results

### 3.1. Demographical and Clinical Data

This study included 178 dogs with Hymenoptera stings out of 19,413 patients admitted to the Small Animal Emergency Clinic and the Clinic for Small Animal Internal Medicine/Dermatology between January 2018 and December 2022. The total incidence was 0.9% or 93 cases per 10,000. The clinical data are presented in Table 3.

The affected dogs had a median age of 1.7 years (range 0.2–14.7 years) with a median weight of 9.85 kg (range 0.8–46 kg). Of the 178 dogs, 107 (60%) were female (55% intact, 45% sterilized), while 71 (40%) were male (69% intact, 31% castrated). A total of 143 out of the 178 (80%) dogs were purebred and 35 (20%) were of mixed breed. The most common breeds were French bulldogs, with eighteen individuals (10%), dachshunds with ten (6%), Labrador retrievers with eight (4%), Yorkshire terriers with seven (4%), Maltese with six (3%), and Spitz, with five (3%). Further affected breeds are listed in Appendix A.

A total of 85 out of 178 (48%) dogs showed symptoms restricted to the skin (grade 1). Seventy-four (41%) showed a combination of skin and gastrointestinal and/or respiratory signs (grade 2), while nineteen (11%) suffered from skin, gastrointestinal/respiratory, and cardiovascular and/or neurological symptoms (grade 3). A total of 56 out of 67 (84%) dogs with a grade 2–3 reaction had no history of previous stings.

2018 was the year with the lowest number of stings with 13 (7%), while 2022 was the year with the highest number of cases with 53 (30%). 2019 had 23 cases (13%), 2020 had 43 cases (24%), and 2021 had 46 cases (26%). Stings were seen almost all year round, with January and December being the only months when no dogs were admitted. July was the month with the most cases, totaling 52 (29%) over the five years. Collectively, most cases happened in summer with 124 out of 178 (69%), then autumn with 28 (16%), followed by spring with 25 (14%), and finally one case (1%) in winter. Of the 166 cases where the time of sting was documented, 67 (40%) were stung at night (6 pm–6 am) and 99 (60%) during the day (6 am–6 pm). Of the 143 cases where the location of the sting was known, 121 (85%) dogs were stung while on a walk, 16 (11%) in the garden, 5 (3%) on the balcony, and 1 (1%) at a dog training center.

For 83 (47%) cases, the Hymenoptera responsible was identified, with 57 (69%) being stung by bees and 26 (31%) stung by wasps. No bumblebee or hornet stings were seen. In 163 (92%) cases, the localization of the insect sting was documented, with the face (63, 39%), paw (46, 28%), and the oral cavity (35, 21%) being the most frequent sites. Additionally, thirteen (8%) dogs were stung in other localizations and six (4%) dogs had multiple stings on different parts of the body. Twenty-six (15%) dogs had experienced a previous sting and six (3%) had been stung twice before. The median time of re-sting was two months with a range from one week to four years. Seven (4%) dogs had been previously diagnosed with atopic dermatitis with six (86%) of them having a cutaneous reaction and only one (14%) having a systemic reaction. Five (3%) dogs were suffering from inflammatory bowel disease (IBD) and seven (4%) had other comorbidities (otitis media, Cushing’s syndrome, hypothyroidism, mastitis, mammary tumor, endocardiosis, a herniated disk).

Gallbladder edema was identified in 22 (28%) of the 80 (45%) dogs who underwent ultrasound examination, while the alanine transaminase (ALT) levels were elevated in 19 (50%) of the 38 (21%) dogs tested.

The mean hospitalization time was 5.2 h, ranging from 1 to 96 h. Dogs with a grade 1 reaction were hospitalized for two hours, while dogs experiencing a grade 2–3 reaction spent 8 h on average in our clinic. Only one dog was not admitted but rather assessed by the emergency veterinarian as not being sufficiently sick for admission (i.e., triaged away). The duration of hospitalization was known in 169 dogs, 111 (65%) dogs were admitted for less than two hours, 35 (21%) dogs for two to six hours, and 23 (14%) dogs for more than six hours. Treatments included antihistamines (154, 89%), glucocorticoids (90, 51%), infusions (63, 36%), gastrointestinal protectants (11, 6%), and epinephrine injections (9, 5%), and 62 (35%) dogs received some other kind of medication. All dogs recovered and were discharged, and eight (4%) underwent venom immunotherapy (VIT). Five of these dogs received a sting provocation, which all of them endured without any adverse effects. Ten (6%) dogs had undergone serological allergy testing with all of them being sensitized to bee venom, eight (80%) also to wasp venom, and five (50%) to hornet venom.

### 3.2. Natural History

Table 4 shows the data regarding the follow-up reactions of 43 dogs (24% of all patients) with subsequent Hymenoptera stings. Twenty-four (56%) of them had only cutaneous symptoms at their first sting and nineteen (44%) had systemic reactions. Fourteen (58%) of the dogs with a mild reaction at the first sting experienced a mild reaction again, and ten (42%) developed systemic episodes. In seven (37%) dogs with first-time moderate to severe reactions, the reaction after re-sting was mild, and moderate to severe in twelve (63%) cases. Sixteen dogs (37%) were stung by bees, ten (23%) by wasps, and in seventeen dogs (40%), the culprit insect could not be definitely identified. The sting occurred 2–12 months after the first one in seventeen patients (40%), in fourteen dogs (32%) within two months, in five dogs (12%) within 1–3 years, and in seven dogs (16%) the interval was unknown. More than half of the stings occurred while the dogs and their owners were on a walk (58%, 25), nine stings (21%) occurred in the garden, and two (5%) on the balcony, with seven unknowns (16%). Seventeen dogs (40%) were stung in the face, eight dogs (18%) on the paw, five dogs (12%) in the oral cavity, and four (9%) somewhere else, with six unknowns (14%). One dog was stung in the paw and the face, another dog was stung in the face and inside the oral cavity, and one dog ingested the insect and was stung in the abdominal region.

### 3.3. Risk Factors

Table 5 shows the results of the multivariable analysis of the ordinal regression model, examining the relationship between the severity of anaphylactic episodes and several predictor variables. The variable “sting in oral cavity” was statistically significant with a *p*-value of 0.01 and an odds ratio (OR) of 2.62 (CI 95% 1.25–5.49). Other variables with a *p*-value < 0.05 were (1) age under two years (OR 2.11, CI 95% 1.13–3.93), (2) weight under 10 kg (OR 2.06, CI 95% 1.11–3.84), (3) purebred dogs (OR 2.43, CI 95% 1.08–5.48), and (4) those that had comorbidities at the moment of the Hymenoptera sting (OR 4.48, CI 95% 1.28–15.71). The results of the bivariate analysis are presented in Appendix A.

### 3.4. Owner Life Quality and State of Awareness for Preventative Measures

For 63 dogs (35%), follow-up was available. The owners of 20 (32%) of these dogs reported HVA having a direct impact on their quality of life. Five (25%) of them said that they were more cautious while on walks, three (15%) stated that they were very afraid of a new sting, and one (5%) person reported that they suffered from chronic stress. Thirteen (21%) owners said that they had changed their lifestyle because of their dog’s disease. Owners reported that twelve (19%) dogs developed new allergies; eight (75%) became allergic to food, and four (25%) to environmental allergens. Eighteen (29%) dogs were said to have a tendency for playing with insects. Eight (13%) owners reported that they owned an adrenaline auto-injector kit for their dogs. All but one of these owners said that the stings had an impact on their quality of life. Awareness of VIT was low, with only 24 (38%) being familiar with the therapy. Furthermore, only four (17%) were convinced of VIT being efficacious, eight (33%) did not believe VIT to be effective, and twelve (50%) were unsure if VIT could prevent future systemic reactions. Thirty owners were tested for their ability to recognize individual Hymenoptera species. Of these thirty owners, twenty-three (77%) correctly identified the species, while seven (23%) identified them incorrectly.

## 4. Discussion

The 0.9% incidence of HVA at our clinic represents a notable increase compared to the 0.12% reported by Rostaher et al. in 2014, where only 24 dogs with cutaneous signs of immediate-type hypersensitivity triggered by different allergens were included [6]. This is partly due to the different inclusion criteria and a larger study population (178 dogs vs. 24 dogs). In our study, 52% of dogs experienced moderate to severe reactions (grade 2–3), a prevalence comparable to those observed in human populations in Europe (0.3% and 0.3%) [42]. Interestingly, 84% (56 of 67) of dogs with grade 2–3 reactions had no prior history of stings, which is not consistent with the human data. This is most likely because many stings in dogs go unnoticed if the owner is not present when the sting occurs.

The rising number of cases (13 in 2018, 53 in 2022) during the study period can partially be attributed to an increase in the number of canine patients at the Vetsuisse Faculty Animal Hospital of the University of Zurich and a 95% increase in new registrations in the Swiss canine population (7243 in 2018, 14,124 in 2022) [43]. Anaphylactic reactions in general are also being seen more frequently due to increasing diagnostic and therapeutic modalities [35]. The true prevalence of Hymenoptera-induced anaphylactic reactions in dogs is likely to be underestimated, as many cases go undiagnosed or are treated in primary practices.

The young median age of 1.7 years aligns with other studies on canine anaphylaxis (1.2 years and 3.2 years) [32,44]. The large population of young dogs in our study may be partly due to the influx of puppies in the total Swiss population due to the COVID-19 pandemic. From February 2020, one month before the World Health Organization (WHO) declared the COVID-19 outbreak as a pandemic, to February 2022, there was a 27% increase in dogs aged 0–1 years (33,507 to 42,553) [43,45]. Another explanation for their overrepresentation could be their playing behavior, which exposes them to stings. Age under two years was also identified as a risk factor for severe reactions while adjusting for other factors, consistent with previous findings that younger dogs are more prone to anaphylaxis [46]. A reason for the more severe reactions in young dogs may be that their immune system is not yet fully developed, as is seen in the higher risk of infectious diseases occurring between birth and the first six months of life [47]. In contrast, studies in humans have shown that older age increases the risk for severe reactions, with every year being associated with a 1.6% increased risk of anaphylaxis [25,48]. Possible explanations encompass the higher exposure of people aged 20 to 45 years to insect stings and the likelihood of older people to suffer from comorbidities like clonal mast cell disease [41]. While we also identified comorbidities as a risk factor for systemic sting reactions, only 14 out of the total 178 dogs (8%) were in this group, and none of our patients were diagnosed with clonal mast cell diseases, which are reported in humans. In contrast to a past study that demonstrated that dogs with canine atopic dermatitis (CAD) were at increased risk of developing urticaria or anaphylaxis (11 of 24 dogs), only one out of seven (14%) dogs with CAD in our study suffered a SSR [6]. Further research is required to establish whether comorbidities are a reliable risk factor for severe anaphylactic reactions.

While more female than male dogs were included in our study, the sex did not have an impact on the severity of anaphylaxis. In the past, male sex was thought to be a risk factor for systemic reactions due to a higher exposure to Hymenoptera stings in humans, which was not demonstrated in our population. Recent studies have shown that anaphylactic episodes are more pronounced in female than male mice due to the female steroid estradiol [49]. Additional studies are needed to determine whether sex is a consistent predictor of systemic sting reactions.

A total of 80% (143) of dogs admitted to this study were purebred, with the most frequent being French bulldogs, dachshunds, Labrador retrievers, Yorkshire terriers, Maltese, and Spitz. This finding is similar to previous studies, where purebreds made up 92% (22 of 24) and 78% (75 of 96) of all dogs [6,32]. In addition to this, 76% of dogs in Switzerland are purebred (424,903 of 556,423, as of 2023), which also explains the distribution in our study [43]. In comparison to the canine population of Switzerland, French Bulldogs (10% vs. 3%), dachshunds (6% vs. 1%), and Yorkshire terriers (4% vs. 3%) were overrepresented [43]. Turner et al. also reported an overrepresentation of dachshunds in their study population compared to a wider hospital sample (6%, 12 of 186, *p* < 0.001) [44]. Dachshunds have been shown to frequently have allergic reactions after vaccination, with up to 36% (31 of 85) of all cases in one study from 2005 [50]. In addition, dachshunds have a higher risk of developing skin diseases, although the exact cause for this is unknown [51]. It was beyond the scope of this study to examine this finding, but further studies investigating this association are warranted. This study also identified that purebred dogs were at a higher risk of developing severe systemic reactions (SSRs) than mixed-breed dogs. Studies that have investigated the prevalence of inherited disorders among mixed-breed and purebred dogs have indicated that some disorders are found in greater prevalence among purebred dogs [52]. Purebred dogs may have a more homogenous genetic pool than mixed dogs, which may impact their immune response to insect stings. Due to the low prevalence of individual breeds, we could not perform the statistical analysis for the individual breed. Showing that pure breed is associated with the disease points toward the possibility of a hereditary risk factor. In humans, hereditary alpha-tryptasemia is a risk factor for Hymenoptera venom allergy [53]. Future research should consider analyzing specific breed characteristics in larger and more diverse populations to assess whether individual dog breeds show distinct patterns of HVA, thus clarifying whether particular breeds have unique predispositions due to genetic or behavioral traits.

Dogs in our study had a median weight of 9.9 kg, ranging from 0.8 to 46 kg, similar to findings by Quantz et al. (13.8 kg, 1.5 to 43.2 kg) and Turner et al. (10.6 kg, 7.3 to 16.9 kg) [32,44]. In line with the hypothesis that smaller animals may have more severe reactions due to a proportionally greater venom exposure per body weight, our data showed that being under 10 kg was a risk factor for SSRs. However, the relationship between the amount of histamine released and the dose of venom in dogs has yet to be fully explained [32].

The last risk factor for SSRs that we identified was a sting inside the oral cavity. As the behavior of catching insects with the mouth is unique to animals, a comparison to humans is not possible. The interaction of venom with the gastrointestinal mucosa may exacerbate the immune response, as suggested by studies on oral venom administration in other species [54]. Additionally, soft tissue swelling and resulting dyspnea are likely key contributors to the severe nature of these reactions.

We gathered information on the natural history of 43 dogs that were stung more than once to determine whether the severity of a previous reaction could predict the outcome of repeated exposure. Of the 24 (56%) dogs with an initially mild reaction, 14 (58%) experienced a mild reaction again, while 10 (42%) developed an SSR. On the other hand, out of the 19 (44%) dogs with first time SSRs, 7 (37%) presented with a mild reaction and 12 (63%) with an SSR at re-sting. These results align with human studies, where adults with a history of SSR have a 40–60% chance of experiencing another episode of the same severity. Additionally, most patients with cutaneous reactions tend to experience the same or less severe level of reaction [3,16,21,22,55]. This high recurrence rate highlights the importance of recommending VIT to prevent further episodes.

Abdominal ultrasound examinations in dogs suspected of anaphylaxis can provide valuable information due to the hepatic venous system being the primary shock organ and the effects of portal hypertension [56,57,58]. Gallbladder wall abnormalities were observed in 27% (22) of dogs undergoing ultrasound, with a higher prevalence (44%, 18) in dogs with SSRs, though this was lower than previous findings [32,44]. This variation could be due to regional differences of insect venom, differences in the severity of anaphylaxis, and the specific technique used to obtain the ultrasound findings [59]. ALT was elevated in 50% (19) of dogs tested, consistent with other studies demonstrating its utility as a marker for anaphylaxis [32,44]. Most owners (77%, 23) were able to correctly identify the different Hymenoptera species, which we tested with pictures. This supports the reliability of owner-reported data on insect encounters and sheds further light on the question of whether the cause for an anaphylactic reaction can be identified by anamnesis alone. Similar to the study conducted by Rostaher et al. in 2023, where 60% (6 of 10) of insect stings were due to bees, the majority of insect triggers in our population were identified as bees (69%, 57), followed by wasps (31%, 26). No bumblebees or hornets were found to have caused stings in this time period. The insect type was not associated with the severity of the clinical reaction in our study population.

Due to the occurrence of biphasic anaphylactic reactions in humans and one beagle, prolonged observation of dogs suffering from anaphylaxis is recommended [60,61]. As expected, dogs with more severe reactions were hospitalized longer than those with mild reactions (8 h vs. 2 h). The 100% survival rate of our population is consistent with other studies [6,44,46].

While adrenaline was used in 7 of 93 (8%) dogs with SSRs, which is an improvement from the study in 2014 by Rostaher et al. where no administration of epinephrine was observed, there is still room for improvement [6]. In Turner et al., 72 out of 232 (31%) dogs suspected of anaphylaxis were treated with adrenaline [44]. There may be a reluctance in the use of adrenaline for anaphylactic shocks because of a fear of adverse effects, even though the intramuscular application of epinephrine shows excellent safety [62]. The high efficacy and safety observed in the VIT administered to the eight patients in our study aligned with data gathered in both canine and human populations [15,16,19]. Additionally, its potential to improve the quality of life underscores its value in managing HVA, which has also been shown in humans. In cases where patients relied solely on adrenaline auto-injectors, the QOL declined over time [12,13]. In this study, all but one of the owners (7 of 8, 88%) who owned auto-injectors reported that insect stings in their dogs had an impact on their QOL. A possible explanation might be that while the auto-injector serves as a constant reminder of the risk of anaphylaxis, a controlled sting provocation after VIT is a direct confirmation of its efficacy. Increased awareness is needed to encourage the use of VIT in veterinary practice. Furthermore, owners of high-risk dogs should be preventively educated on how to respond to insect stings and their potential consequences.

Several possible limitations of this research should be considered when interpreting the findings. Due to the retrospective nature, we had to rely on existing records that might not have always accurately captured all of the relevant details. The use of a questionnaire introduces the possibility of recall and non-response bias. Finally, the relatively low number of recurrent stings limits the generalizability of our findings on the natural history of Hymenoptera venom allergy in dogs.

## 5. Conclusions

This study provides novel insights into the natural history and risk factors of dogs stung by Hymenoptera, being the first of its kind in veterinary medicine. We revealed that dogs under two years, weighing less than 10 kg, of purebred status, and those being stung inside the oral cavity are at a higher risk of experiencing SSRs after Hymenoptera stings. A significant proportion of dogs develop similar or worse reactions upon subsequent stings, emphasizing the importance of VIT to prevent such occurrences.

## Figures and Tables

**Table 1 animals-14-03220-t001:** Differential diagnoses to symptoms caused by Hymenoptera stings (adapted from Rostaher et al. [6]).

Symptoms	Differential Diagnoses
Wheals	Bacterial folliculitis
	Vasculitis
	Erythema multiforme
	Cutaneous lymphoma
	Mastocytosis
Angioedema	Juvenile/infectious cellulitis
	Mast cell tumor
	Cutaneous lymphoma
	Lymphoedema
Anaphylaxis	Vasodepressor events
	Systemic mastocytosis
	Pheochromocytoma
	Shock

**Table 2 animals-14-03220-t002:** Clinical grading system for anaphylaxis (adapted from Rostaher et al. [20]).

Grade	Organ System Involved	Clinical Findings
1—Mild	Skin	Generalized erythema, urticaria and/or angioedema
2—Moderate	Gastrointestinal, respiratory +/− skin	Dyspnea, stridor, wheeze, nausea, vomiting and/or abdominal pain
3—Severe	Cardiovascular, neurological +/− gastrointestinal, respiratory +/− skin	Cyanosis, pallor, SpO2 < 92%, hypotension *, collapse, loss of consciousness, incontinence

* Hypotension was defined as systolic blood pressure < 90 mmHg.

**Table 3 animals-14-03220-t003:** Demographics of 178 dogs stung by Hymenoptera.

Variable	Parameter
Age	
Median age in years (min., max.)	1.7 (0.2, 14.7)
Age range, no. (%)	
0–2 years	100 (56)
2–7 years	49 (28)
>7 years	29 (16)
Median weight in kg (min., max.)	9.9 (0.8, 46)
Sex and reproductive status, no. (%)	
Female intact	59 (33)
Female sterilized	48 (27)
Male intact	49 (28)
Male castrated	22 (12)
Breed, no. (%)	
Purebred	143 (80)
Mixed	35 (20)
French bulldog	18 (10)
Dachshund	10 (6)
Labrador retriever	8 (4)
Yorkshire terrier	7 (4)
Maltese	6 (3)
Spitz	5 (3)

**Table 4 animals-14-03220-t004:** Follow-up reactions in 43 cases with subsequent Hymenoptera stings (grades 1 vs. 2–3).

Reaction on First Sting	Reaction on Follow-Up Sting	
	Grade 1	Grade 2–3
Grade 1	14 (58%)	10 (42%)
Grade 2–3	7 (37%)	12 (63%)

**Table 5 animals-14-03220-t005:** Multivariable ordinal regression analysis of factors associated with severe anaphylaxis.

Variables	Odds Ratio	CI 95%	*p*-Value
Age < 2 years	2.11	1.13–3.93	0.02 *
Weight < 10 kg	2.06	1.11–3.84	0.02 *
Male sex	0.97	0.52–1.81	0.92
Purebred	2.43	1.08–5.48	0.03 *
Comorbidity	4.48	1.28–15.71	0.02 *
>1 episode	1.10	0.44–2.72	0.84
Sting in oral cavity	2.62	1.25–5.49	0.01 *
Bee venom	1.27	0.65–2.48	0.48

* Statistically significant.

## Data Availability

All data necessary to replicate this analysis are contained within this article.

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
