# Peer review of "Natural History and Risk Factors of Hymenoptera Venom Allergy in Dogsâ€"

_animals, 2024, doi:10.3390/ani14223220_

Round 1
Reviewer 1 Report
Comments and Suggestions for Authors
Reviewer Remarks to “Natural history and risk factors of Hymenoptera venom allergy in dogs”
The authors focused on the occurrence of HVA, sometimes leading to SSR, in dogs. Research compared retrospective data from dogs, with data from human occurrence, hypothesizing that the prevalence, as well as risk factors for SSR in dogs could resemble what is commonly observed in humans. After a clear discussion of the results, at the end, authors present a set of conclusions as relevant risk factors and emphasize the importance of VIT to prevent the worsening of clinical manifestations associated to successive stings.
Remarks:
In the first line of the third paragraph of page 2: correct “…huzzmans…” to “…humans…”.
In the Materials and Methods section, when the authors use “daytime” in the paragraph before “3. Results”, do they mean during the day, as opposed to by night, or do they mean time of the day? It seems to be time of the day, by the results presented. Please correct accordingly.
Four lines below, in the same paragraph, what do the authors intend with “…bee venom and comorbidities.”, and please correct “comorobidities” to “comorbidities”.
In the abstract, the authors refer “…178 dogs that were stung by Hymenoptera between 2018 and 2022.”. However, in the 1st paragraph of the results, the authors mention “This study included 178 cases of Hymenoptera stings…”. The same figure of 178 dogs as well as stings is just a coincidence? In fact, there were dogs stung more than once (43 dogs – as mentioned in the first line of page 9).
In the section “Sex and reproductive status, no. (%)” of table 3, please align the figures on the right towards the respective items on the left.
In the first three paragraphs after table 3 (but also along the text), there are sentences starting by numerals. It should be rearranged to avoid that.
In the first paragraph after table 3, instead of “The most common breeds were French bulldogs (18, 10%), dachshunds (10, 6%), Labrador retrievers (8, 4%), Yorkshire terriers (7, 4%), Maltese (6, 3%) and spitz (5, 3%).” I would suggest “The most common breeds were French bulldogs, with 18 individuals (10%), dachshunds, with 10 (6%), Labrador retrievers, with eight (4%), Yorkshire terriers, with seven (4%), Maltese, with six (3%) and Spitz, with five (3%).” as it will keep the same format used in the table and in the following paragraph, with just the percentage figures in parentheses. It sounds clearer to the reader.
Fourth to last line of the last paragraph of page 7: correct “…there was an 27% increase in dogs aged 0-1 years…” to “…there was a 27% increase in dogs aged 0-1 years…”.
In the last sentence of page 8 the authors somehow postulate that soft tissue swelling with subsequent dyspnea should be considered less significant as only three dogs that have been stung in the mouth, developed respiratory distress. However, those three dogs were (most probably) the majority of this subgroup of five dogs (12%) that have been stung in the oral cavity. Furthermore, the authors affirm in the conclusions section that dogs being stung in the oral cavity are at a higher risk of experiencing SSR. I would suggest reformulating the sentence regarding this fact and the reduced number of dogs that have been stung in the oral cavity.
Second paragraph of page 9: “…[32,44] This…” to “…[32,44]. This…”.
Author Response
Review 1:
In the first line of the third paragraph of page 2: correct “…huzzmans…” to “…humans…”.
- Thank you for the notice, we have corrected this typo
In the Materials and Methods section, when the authors use “daytime” in the paragraph before “3. Results”, do they mean during the day, as opposed to by night, or do they mean time of the day? It seems to be time of the day, by the results presented. Please correct accordingly.
- Thank you for the advice, we have changed it accordingly
Four lines below, in the same paragraph, what do the authors intend with “…bee venom and comorbidities.”, and please correct “comorobidities” to “comorbidities”.
- We have corrected the typo. We have also inserted a comma between “bee venom” and “comorbidities” to make clearer that these two variables are separate (so bee venom is a variable compared to other types of venom and comorbidities is compared to having no comorbidities).
In the abstract, the authors refer “…178 dogs that were stung by Hymenoptera between 2018 and 2022.”. However, in the 1st paragraph of the results, the authors mention “This study included 178 cases of Hymenoptera stings…”. The same figure of 178 dogs as well as stings is just a coincidence? In fact, there were dogs stung more than once (43 dogs – as mentioned in the first line of page 9).
- Thank you for pointing this out. We agree that it isn’t correct, so we have changed the wording to “178 dogs with Hymenoptera stings”
In the section “Sex and reproductive status, no. (%)” of table 3, please align the figures on the right towards the respective items on the left.
- Thank you for the notice. In our word version of the manuscript the figures and text are aligned but in the PDF it appears like you describe. We don’t exactly know how to make sure there aren’t any changes in the formatting when the document is made in a PDF.
In the first three paragraphs after table 3 (but also along the text), there are sentences starting by numerals. It should be rearranged to avoid that.
- Thank you for your comment, I agree that it doesn’t read as nice, so we have changed it to the following text: “The affected dogs had a median age of 1.7 years (range 0.2-14.7 years) with a median weight of 9.85 kg (range 0.8-46 kg). Of the 178 dogs, 107 (60%) were female (55% intact, 45% sterilized), while 71 (40%) were male (69% intact, 31% castrated). 143 out of the 178 (80%) dogs were purebred and 35 (20%) were of mixed breed.”
In the first paragraph after table 3, instead of “The most common breeds were French bulldogs (18, 10%), dachshunds (10, 6%), Labrador retrievers (8, 4%), Yorkshire terriers (7, 4%), Maltese (6, 3%) and spitz (5, 3%).” I would suggest “The most common breeds were French bulldogs, with 18 individuals (10%), dachshunds, with 10 (6%), Labrador retrievers, with eight (4%), Yorkshire terriers, with seven (4%), Maltese, with six (3%) and Spitz, with five (3%).” as it will keep the same format used in the table and in the following paragraph, with just the percentage figures in parentheses. It sounds clearer to the reader.
- Thank you for your comment, I agree that it doesn’t read as nice, so we have changed it to the following text: “The most common breeds were French bulldogs, with 18 individuals (10%), dachshunds with ten (6%), Labrador retrievers with eight (4%), Yorkshire terriers with seven (4%), Maltese with six (3%) and Spitz, with 5 (3%).”
Fourth to last line of the last paragraph of page 7: correct “…there was an 27% increase in dogs aged 0-1 years…” to “…there was a 27% increase in dogs aged 0-1 years…”.
- Thank you for the notice, we have corrected this typo
In the last sentence of page 8 the authors somehow postulate that soft tissue swelling with subsequent dyspnea should be considered less significant as only three dogs that have been stung in the mouth, developed respiratory distress. However, those three dogs were (most probably) the majority of this subgroup of five dogs (12%) that have been stung in the oral cavity. Furthermore, the authors affirm in the conclusions section that dogs being stung in the oral cavity are at a higher risk of experiencing SSR. I would suggest reformulating the sentence regarding this fact and the reduced number of dogs that have been stung in the oral cavity.
- That is a point that we did not consider, thank you very much. We have changed it into the following text: “Additionally, soft tissue swelling and resulting dyspnea could be likely key contributors to the severe nature of these reactions”.
Second paragraph of page 9: “…[32,44] This…” to “…[32,44]. This…”.
- Thank you for the notice, we have corrected this typo
Reviewer 2 Report
Comments and Suggestions for Authors
Edwin Chapman et al. „Natural history and risk factors of Hymenoptera venom allergy in dogs“.
The submitted manuscript is written clearly, distinctly and is in my view a significant contribution, especially for practicing veterinarians, as there is little information on allergic reactions of dogs after bites by Hymenoptera insects. The authors have evaluated and assessed all the factors studied in allergic reactions after Hymenoptera bites, used the correct methodologies and statistical evaluation of the results, and the assessment and description of medication is also beneficial.
I have the following minor comments and recommendations on the manuscript:
1. Page 2 - second paragraph from top - first sentence - technical error, should correctly be ... in humans... and not as stated ...in huzzmans..."
2. Please modify Table 3, under Sex and reproductive status, no (%) - there is a shift in the numbers of the results in the right part of the table, the figure for Male intact is missing (I assume this is a technical error); I also recommend to add to this table the results that the authors describe in the text for the Spitz breed.
After minor modifications, I recommend to publish the manuscript in the journal Animals.
Author Response
Review 2:
- Page 2 - second paragraph from top - first sentence - technical error, should correctly be ... in humans... and not as stated ...in huzzmans..."
- Thank you for the notice, we have corrected this typo
- Please modify Table 3, under Sex and reproductive status, no (%) - there is a shift in the numbers of the results in the right part of the table, the figure for Male intact is missing (I assume this is a technical error); I also recommend to add to this table the results that the authors describe in the text for the Spitz breed.
- Thank you for the notice. We have included the spitz breed in the table. In my word version of the manuscript the figures and text are aligned and complete but in the PDF it appears like you describe. I don’t exactly know how to make sure there aren’t any changes in the formatting when the document is made in a PDF.
Reviewer 3 Report
Comments and Suggestions for Authors
General remarks
The manuscript is well written, with a clear structure, clearly written material and methods and presentation of results, with a structured discussion and clearly understandable conclusions. The results represent the first data on the natural history of dogs that developed severe systemic reactions after bee or wasp stings that have not been previously analysed in any veterinary study. In addition, for the first time in veterinary science, risk factors for the development of anaphylaxis in dogs have been identified and statistically demonstrated.the topic is original and relevant to the field as it fills a specific gap in the field. The research is based on methodology and number of animals sufficient to support the conclusions. The conclusions are consistent with the evidence and arguments presented and address the main question posed. The tables and figures are representative.
References are appropriate but misquoted at one point in the manuscript (see below).
Minor notes:
Page 2, second paragraph - overall: the wrong literature is cited here when talking about animal patients, but human studies are cited. The only veterinary studies that need to be cited here are numbers 6, 19 and 20 (Rostaher et al., 2017; Ewing et al., 2022 and Rostaher et al., 2023 respectively). The citations numbered 7-18 should not have been cited in this paragraph, except for the last statement »Human studies report efficacy of 80-95%, with similar results published in dogs undergoing VIT [14-20].«
The other way to reconstruct this paragraph is to delete the part of the first sentence of this paragraph, namely »If a patient suffers a severe reaction and the owner is motivated to do so, detection of venom-specific IgE antibodies can be performed by blood tests and skin tests..." is deleted so that only humans are mentioned.
Page 2, third paragraph: »Studies to investigate the natural history of HVA in huzzmans«, the letters zz should be deleted.
Author Response
Review 3:
Page 2, second paragraph - overall: the wrong literature is cited here when talking about animal patients, but human studies are cited. The only veterinary studies that need to be cited here are numbers 6, 19 and 20 (Rostaher et al., 2017; Ewing et al., 2022 and Rostaher et al., 2023 respectively). The citations numbered 7-18 should not have been cited in this paragraph, except for the last statement »Human studies report efficacy of 80-95%, with similar results published in dogs undergoing VIT [14-20].« The other way to reconstruct this paragraph is to delete the part of the first sentence of this paragraph, namely »If a patient suffers a severe reaction and the owner is motivated to do so, detection of venom-specific IgE antibodies can be performed by blood tests and skin tests..." is deleted so that only humans are mentioned.
- Thank you for checking the citations. We agree that I have wrongly stated human studies, and it reads as though the information applies to dogs. We have changed this part of the text so only the last sentence applies directly to canine patients, as you advised: “To date, diagnosis of a severe reaction to Hymenoptera venom is achieved by a thorough anamnesis and observation of a typical clinical pattern [8]. Treatment of a systemic allergic reaction is based on the severity of the clinical signs. While mild reactions involving only cutaneous symptoms can be treated with antihistamines and glucocorticoids, the first-line treatment for an anaphylactic shock is epinephrine [9]. If a patient suffers a severe reaction, detection of venom specific IgE antibodies through blood work and skin tests may be performed to discover or confirm the culprit allergen and to build a basis on which venom immunotherapy (VIT) may be carried out [10]. VIT is the only treatment shown to reliably prevent further anaphylactic episodes and improve quality of life (QOL) compared to those relying solely on adrenaline auto-injectors [11-13]. Human studies report an efficacy of 80-95% with similar results published in dogs undergoing VIT [14-20].”
Page 2, third paragraph: »Studies to investigate the natural history of HVA in huzzmans«, the letters zz should be deleted.
- Thank you for the notice, we have corrected this typo
Reviewer 4 Report
Comments and Suggestions for Authors
This is a very well-written paper that will serve as a valuable contribution to canine health and veterinary medicine. I do have one major concern about the treatment and interpretation of the 'purebred dogs' category. This and other minor edits are detailed in the attached.
This paper is very well-written and easy to follow. I think it is an excellent contribution, but I do have one major concern I would like to see addressed, which regards the way purebred and mixed breed status of the dogs was considered.
General considerations
Main concern:
I believe that the lumping of ‘purebred dogs’ into a single category is a major shortcoming. It is also difficult to distinguish between the purebred and mixed breed contributions - in the methods, discussion and as presented in Table 3.
Physiology and related concerns aside, there are large behavioural differences between the main types of dogs listed, which could predispose or insulate these types of dogs to insect stings. French bulldogs, dachshunds, Labrador retrievers, Yorkshire terriers, Maltese and spitz and then whatever mixed breed dogs presented –I would like to see if the authors can at least try to analyze or examine whether there were any differences or patterns on the basis of type of dog.
Even if anecdotally, this could be quite relevant and could be incorporated into recommendations for followup studies on the basis of types of dogs perhaps more prone to insect stings.
(In parallel there could also be certain behaviours and habits in the owners in association with the types of dogs, though this is of course well outside the scope of the study).
Out of curiosity…
How easy and cost-effective might it be to routinely monitor dogs for insect stings, say on a yearly basis as part of annual exam? Would this be worth doing for all, or perhaps certain dogs having shown either a vulnerability or tendancy towards being bitten?
Specific comments
In abstract – please identify the location of the study
Keywords – don’t use same words as in title
On page 8 it says:
‘This study also identified that purebred dogs were at a higher risk of developing severe systemic reactions (SSR) than mixed-breed dogs.’
As mentioned and explained above, I believe that the lumping of ‘purebred dogs’ into a single category is a drawback of the analysis.
To the statement --
‘Almost two thirds of patients with SSR experienced the same or worse symptoms after subsequent stings and > 40% of patients with local reactions developed SSR when stung again.’
Were any patterns seen that could point to the dogs most likely to be stung again?
What is actually known about VIT as a preventive for dogs? Are there any downsides? It would be relevant to add a sentence or two in the introduction.
Introduction --
First paragraph - spell out Immunoglobulin E (IgE)
-Any relevance about dogs and humans being stung together? Also, relative to other domesticated animals a small- or large-animal practice veterinarian might see, are dogs the most frequently seen patient with hymenoptera stings?
Page 2 – humans, not huzzmans
Last paragraph – should it not be ‘life quality of the canine patient’ versus life quality of the owner? (Same question for page 4)
Then, when we come to Section 3.4 I see why it is framed as ‘life quality of the owner’. I this is not an accurate term. I suggest changing to something more like ‘state of awareness or vigilance of the owner’ to better reflect how they adapted the sting to the dog and awareness into their usual routine.
Page 3 – Section 2.2 - Data collection - ‘time of day’ of sting?
Section 2.3 – sex, not gender
This reads repetitive:
‘The variables assessed in the bivariate analysis were: age under two years, weight under 10 kg, male sex, purebred dogs, more than one episode, sting interval under two months, a previous systemic reaction, daytime, sting in the oral cavity, bee venom, canine atopic dermatitis and comorbidities. Since we had sufficient data for these variables and a strong clinical rational, we included the following variables in the multiviariable analysis: age under two years, weight under 10 kg, male sex, purebred dogs, more than one episode, sting in the oral cavity, bee venom and comorobidities.’
10,000 not 10’000, same for other numbers
Table 3 – purebred what, though? What breed specifically?
Why is spitz not in Table 3? Which are the mixed breed participants?
Page 5 – this could be a bit confusing to readers:
‘Of the 166 cases where the time of sting was documented, 67 (40%) were stung between 6pm and 6am and 99 (60%) between 6am and 6pm.’
Can the authors add a disclaimer about evening hours versus daylight hours for utmost clarity?
Stung by honey bees if not by bumble bees? Are members of the public generally able to distinguish between the two? And is it important for the substance of the paper to distinguish between a sting from a honey bee versus from a bumble bee?
Later, on page 6, it says: Ten (6%) dogs had undergone serological allergy testing with all of them being sensitized to bee venom, eight (80%) also to wasp venom and five (50%) to hornet venom.
General bee?
Page 6 – Only one dog was not admitted but triaged away (grade 1). Triaged right away? Is there a word missing?
Section 3.2 - ‘…and one dog ingested the insect and was stung somewhere else.’ Where, then, was the dog stung?
Page 9 - ‘It can be assumed…’ please reword to ‘There may be a reluctance…’
Second to last paragraph:
‘Increased awareness and education are needed to encourage the use of VIT in veterinary practice.’
Authors might also consider it appropriate to add a sentence here about preventively educating new owners and owners of puppies or dogs especially senstive to insect bites about the dangers of this.
Author Response
Review 4:
General considerations
Main concern:
I believe that the lumping of ‘purebred dogs’ into a single category is a major shortcoming. It is also difficult to distinguish between the purebred and mixed breed contributions - in the methods, discussion and as presented in Table 3.
Physiology and related concerns aside, there are large behavioural differences between the main types of dogs listed, which could predispose or insulate these types of dogs to insect stings. French bulldogs, dachshunds, Labrador retrievers, Yorkshire terriers, Maltese and spitz and then whatever mixed breed dogs presented –I would like to see if the authors can at least try to analyze or examine whether there were any differences or patterns on the basis of type of dog.
Even if anecdotally, this could be quite relevant and could be incorporated into recommendations for followup studies on the basis of types of dogs perhaps more prone to insect stings.
(In parallel there could also be certain behaviours and habits in the owners in association with the types of dogs, though this is of course well outside the scope of the study).
- Thank you for this observation. We acknowledge that lumping purebred dogs into a single category may obscure finer differences across specific breeds. However, our study's aim was to assess broad risk factors, and grouping purebred versus mixed-breed dogs (also known as “mutts” or mongrels) was intended to identify general trends rather than breed-specific traits. The following points support the argument that although purebred status and behaviors like insect play may correlate with a risk for severe systemic reactions, exposure is more related to universal dog activities and physiology rather than specific breed behavior traits:
- Given that 85% of stings occurred during outdoor activities, such as walks, it implies that most dogs, regardless of breed, were exposed to stings in similar environments and situations, suggesting this risk factor is likely universal rather than breed specific
- We noted that 29% of owners reported their dogs actively played with insects, a behavior seen across breeds rather than specific to any one type (one might think that hunting dogs like Labrador retrievers or dachshunds might do this more, but this was not confirmed).
- In consideration of this, we have added the following statement in the end of the third paragraph of page 8: “Future research should consider analyzing specific breed characteristics in larger and more diverse populations to assess whether individual dog breeds show distinct patterns of HVA, thus clarifying if particular breeds have unique predispositions due to genetic or behavioral traits.”
- We have also written something on mixed breeds vs. purebred dogs on page 7 of this review.
Out of curiosity…
How easy and cost-effective might it be to routinely monitor dogs for insect stings, say on a yearly basis as part of annual exam? Would this be worth doing for all, or perhaps certain dogs having shown either a vulnerability or tendancy towards being bitten?
- For the cases of most healthy dogs which are stung by Hymenoptera insects, the owner does not observe the sting. This is why dogs are sensitized to Hymenoptera allergens, but there is no clear history of stings. This is the drawback in veterinary medicine, as our patients are not reporting to us. Your idea sounds interesting; however, one has to consider that the risk of being bitten/stung by an insect is not equivalent to the risk of developing anaphylaxis. The serological allergy test that we use in our clinic costs around 250 Swiss Francs and is therefore quite expensive to be part of an annual exam. We are currently thinking of ways to study the prevalence of HVA in the Swiss dog population and it would be great to implement your idea into that study which should have a much higher power.
Specific comments
In abstract – please identify the location of the study
- Thank you for the notice, we have added the following part to the abstract: “This was achieved with an inquiry into the case histories of 178 dogs that were stung by Hymenoptera and presented to the Vetsuisse Faculty Animal Hospital of the University of Zurich between 2018 and 2022.”
Keywords – don’t use same words as in title
- Thank you for noticing this, we have changed it to only contain the following keywords: dog; anaphylaxis; natural history; venom immunotherapy
On page 8 it says:
‘This study also identified that purebred dogs were at a higher risk of developing severe systemic reactions (SSR) than mixed-breed dogs.’
As mentioned and explained above, I believe that the lumping of ‘purebred dogs’ into a single category is a drawback of the analysis.
- Unfortunately, due to the low prevalence of individual breeds we could not perform the statistical analysis for the individual breed. Showing that pure breed is associated with the disease points toward the possibility of a hereditary risk factor. In humans, hereditary alpha-tryptasemia is a risk factor for Hymenoptera venom allergy.
To the statement --
‘Almost two thirds of patients with SSR experienced the same or worse symptoms after subsequent stings and > 40% of patients with local reactions developed SSR when stung again.’
Were any patterns seen that could point to the dogs most likely to be stung again?
- No, sadly we could not identify any significant patterns. The group of dogs in our study that was stung more than once was on average younger (1.4 vs. 3.2 years), had a larger share of male (53% vs. 40%), intact (71% vs. 40%) and purebred (82% vs. 80%) dogs compared to the dogs that had been stung only once. Due to the small sample size of these dogs and because these tendencies were not significant, we chose not to include them in our study. Unfortunately, dogs are exposed to the outdoors on a daily basis, so Hymenoptera stings cannot be prevented. Therefore, we recommend VIT. Also in humans, although they can control their behavior and start avoiding areas known to be inhabited by Hymenoptera insects, the VIT is still recommended.
What is actually known about VIT as a preventive for dogs? Are there any downsides? It would be relevant to add a sentence or two in the introduction.
- Thank you for your comment. We mention VIT in the introduction and reference an article from our study group where we could show the lack of adverse events while performing VIT. As recommended by you, we have now added a sentence that the side effects are rare: “Human studies report an efficacy of 80-95% and few side effects, with similar results published in dogs undergoing VIT [14-20].”
Introduction --
First paragraph - spell out Immunoglobulin E (IgE)
- Thank you for noticing this, we have corrected it in the text
-Any relevance about dogs and humans being stung together? Also, relative to other domesticated animals a small- or large-animal practice veterinarian might see, are dogs the most frequently seen patient with hymenoptera stings?
- Only in 1 case of our dogs, the owner was also stung. In human epidemiologic studies, there was not an association of animal companionship and increased risk of stings. When we started this study, we thought about also including cats but we found that there were very few cases and in these cases, a Hymenoptera stings was mostly suspected but not clearly seen.
Page 2 – humans, not huzzmans
- Thank you for the notice, we have corrected this typo
Last paragraph – should it not be ‘life quality of the canine patient’ versus life quality of the owner? (Same question for page 4)
- The animals cannot report on their life quality , only the owners so the quality of life can only apply to them. This is different if we take atopic dermatitis, a chronic allergic disease affecting the dog’s daily life. Here you can ask the owners to give a subjective evaluation for the life quality of their pets (difficulties sleeping, depressed…). After an anaphylactic episode the dogs do not show such signs, and it seems that their life quality is not affected. They even continue to play with the insects, which is also speaking against the fact that the last HVA event was traumatic for them.
Then, when we come to Section 3.4 I see why it is framed as ‘life quality of the owner’. I this is not an accurate term. I suggest changing to something more like ‘state of awareness or vigilance of the owner’ to better reflect how they adapted the sting to the dog and awareness into their usual routine.
- you are right, this section encompasses more than life quality of the owner. Owner life quality and state of awarness for preventative measures. Please see the comment above why we cannot measure pet life quality,
Page 3 – Section 2.2 - Data collection - ‘time of day’ of sting?
- We have changed it to: “time of day of the sting”. This should be clearer to the readers.
Section 2.3 – sex, not gender
- Thank you for noticing this, we have corrected it
This reads repetitive:
‘The variables assessed in the bivariate analysis were: age under two years, weight under 10 kg, male sex, purebred dogs, more than one episode, sting interval under two months, a previous systemic reaction, daytime, sting in the oral cavity, bee venom, canine atopic dermatitis and comorbidities. Since we had sufficient data for these variables and a strong clinical rational, we included the following variables in the multiviariable analysis: age under two years, weight under 10 kg, male sex, purebred dogs, more than one episode, sting in the oral cavity, bee venom and comorobidities.’
- Thank you, we re-phrased this to the following: “The variables assessed in the bivariate analysis were: age under two years, weight under 10 kg, male sex, purebred dogs, more than one episode, sting interval under two months, a previous systemic reaction, time of day, sting in the oral cavity, bee venom, canine atopic dermatitis and comorbidities. Variables with clinical relevance and sufficient data were further assessed in the multivariable analysis, including age under two years, weight under 10 kg, male sex, purebred status, multiple episodes, sting in the oral cavity, bee venom, and comorbidities.”
10,000 not 10’000, same for other numbers
- Thank you for noticing, we have changed this in the text
Table 3 – purebred what, though? What breed specifically? Why is spitz not in Table 3? Which are the mixed breed participants?
- We have added the Spitz breed to table 3, thank you for noticing that it was missing. We referred to “mixed breeds” for dogs with parents of different breeds. This contrasts with a "purebred" dog, which has parents and ancestors from the same breed and meets specific physical and behavioral characteristics defined by that breed's standard. Purebred dogs are often registered with breed clubs, while mixed breeds have a combination of traits from multiple breeds, resulting in unique appearances and personalities that don’t conform to a single breed standard. Another word for a mixed breed dogs would be a “mutt” or mongrel. We hope this clears up the difference between purebred and mixed dogs!
Page 5 – this could be a bit confusing to readers: ‘Of the 166 cases where the time of sting was documented, 67 (40%) were stung between 6pm and 6am and 99 (60%) between 6am and 6pm.’ Can the authors add a disclaimer about evening hours versus daylight hours for utmost clarity?
- Thank you for the comments, we now corrected the sentence to the following: “Of the 166 cases where the time of sting was documented, 67 (40%) were stung at night (6 pm – 6 am) and 99 (60%) during the day (6 am – 6 pm).”
Stung by honey bees if not by bumble bees? Are members of the public generally able to distinguish between the two? And is it important for the substance of the paper to distinguish between a sting from a honey bee versus from a bumble bee?
- The owners of the animals were asked based on images which insect they observed, they could discern well a bumble bee from a bee, also a bee from a wasp. We added this piece of information now in the materials and methods and the supplementary table S2: “Owners were contacted to evaluate the HVA natural history, assess their quality of life, provide guidance on preventative measures and to test their ability to recognize the different types of Hymenoptera.”
- And also added the results related to this question to the results section (3.4, last sentence): “Thirty owners were tested for their ability to recognize individual Hymenoptera species. Of these 30 owners, 23 (77%) correctly identified the species, while 7 (23%) identified them incorrectly.”
- And have added a part to the discussion (page 9, third paragraph): “Most owners (77%, 23) were able to correctly identify the different Hymenoptera species, which we tested with pictures. This supports the reliability of owner-reported data on insect encounters and sheds further light on the question if the cause for an anaphylactic reaction can be identified by anamnesis alone.
Later, on page 6, it says: Ten (6%) dogs had undergone serological allergy testing with all of them being sensitized to bee venom, eight (80%) also to wasp venom and five (50%) to hornet venom. General bee?
- On page 9 (end of third paragraph), we have added the following text to present our findings clearer: “Similar to the study done by Rostaher et al. in 2023, where 60% (6 of 10) of insect stings were due to bees, the majority of insect triggers in our population were identified as bees (69%, 57), followed by wasps (31%, 26). No bumblebees or hornets were found to have caused stings in this time period. The insect type was not associated with the severity of the clinical reaction in our study population.”
Page 6 – Only one dog was not admitted but triaged away (grade 1). Triaged right away? Is there a word missing?
- “Triage away” is a term used in veterinary medicine, when the emergency veterinarian assesses the patient as not sick enough for an admission to the hospital. In other words, the patient is deemed not to need immediate, intensive medical intervention and is instead given advice or treatment recommendations to follow outside the hospital setting. Here is a reference explaining this process: https://pubmed.ncbi.nlm.nih.gov/1586426/
Section 3.2 - ‘…and one dog ingested the insect and was stung somewhere else.’ Where, then, was the dog stung?
- We have included the exact location of the sting: “One dog was stung in the paw and the face, another dog was stung in the face and inside the oral cavity and one dog ingested the insect and was stung in the abdominal region.”
Page 9 - ‘It can be assumed…’ please reword to ‘There may be a reluctance…’
- Thank you for this comment. We corrected the sentence accordingly: “There may be a reluctance in the use of adrenaline for anaphylactic shocks because of a fear of adverse effects, even though intramuscular application of epinephrine shows excellent safety [61].”
Second to last paragraph:
‘Increased awareness and education are needed to encourage the use of VIT in veterinary practice.’
Authors might also consider it appropriate to add a sentence here about preventively educating new owners and owners of puppies or dogs especially senstive to insect bites about the dangers of this.
- Excellent point, thank you. We have added the following sentence: “Increased awareness is needed to encourage the use of VIT in veterinary practice. Furthermore, owners of high-risk dogs should be preventively educated on how to respond to insect stings and their potential consequences. “
Round 2
Reviewer 4 Report
Comments and Suggestions for Authors
I am very satisfied with the author response in the cover letter and thank them for the consideration of taking the time to provide interesting and stimulating replies.
Now, only a few minor questions and recommendations remain – mainly to enhance clarity and transparency for the reader:
I am still a bit unclear, looking at the layout in Table 3. On the one hand, it is very helpful to have contrasted the total number of purebred versus mixed breed dogs. But how do the numbers reported for purebred dogs pertain to those reported for the 6 most common breeds? For full disclosure, and even if the numbers are tiny, it would be informative to include in the table ALL the breeds seen, even if in the text only the 6 main ones are alluded to. As an example, someone wishing to build on this very important work might be interested in a breed that was in the least seen category and having that referenced in Table 3 would provide context for them.
Or, if the additions of the less seen breeds would be too cumbersome, it would be nice to have the full list of purebred types seen in the supplementary section, which could then be referred to along with Table 3.
Regarding the designation of mixed breed versus purebred – in North America, we also affectionately call mixed breed dogs mutts. I realized after reading the author response that what I was actually unclear about is how dogs were assigned to this category by the authors for this study as being purebred or of mixed breed. The answer might be in this sentence on lines 134 to 135:
'The breed, sex and age of the included population were compared to data from the Swiss dog’s registration system (ANIS) and institutional records.'
Did the owners disclose that the dogs were of mixed race or purebred?
Next, on page 6 I had asked: Only one dog was not admitted but triaged away (grade 1). Triaged right away? Is there a word missing?
- “Triage away” is a term used in veterinary medicine, when the emergency veterinarian assesses the patient as not sick enough for an admission to the hospital. In other words, the patient is deemed not to need immediate, intensive medical intervention and is instead given advice or treatment recommendations to follow outside the hospital setting. Here is a reference explaining this process: https://pubmed.ncbi.nlm.nih.gov/1586426/
Thank you for clarifying. I work closely with veterinarians here in North America and was unfamiliar with the term, so I recommend rewording to:
‘Only one dog was not admitted but rather assessed by the emergency veterinarian as not being sufficiently sick for admission (i.e., triaged away)’
This is a bit longer, but ensures all readers are clear on the circumstances for this dog.
Finally, I greatly appreciate the caveat language added by the authors on lines 328-331. Would they consider creating a new paragraph, including the additional language they provided to me in their cover letter? Something like:
'Purebred dogs may have a more homogenous genetic pool than mixed dogs which may impact their immune response to insect stings. Due to the low prevalence of individual breeds we could not perform the statistical analysis for the individual breed. Showing that pure breed is associated with the disease points toward the possibility of a hereditary risk factor. In humans, hereditary alpha-tryptasemia is a risk factor for Hymenoptera venom allergy. Future research should consider analyzing specific breed characteristics in larger and more diverse populations to assess whether individual dog breeds show distinct patterns of HVA, thus clarifying if particular breeds have unique predispositions due to genetic or behavioral traits.'
Author Response
Comment:
I am very satisfied with the author response in the cover letter and thank them for the consideration of taking the time to provide interesting and stimulating replies.
Now, only a few minor questions and recommendations remain – mainly to enhance clarity and transparency for the reader:
I am still a bit unclear, looking at the layout in Table 3. On the one hand, it is very helpful to have contrasted the total number of purebred versus mixed breed dogs. But how do the numbers reported for purebred dogs pertain to those reported for the 6 most common breeds? For full disclosure, and even if the numbers are tiny, it would be informative to include in the table ALL the breeds seen, even if in the text only the 6 main ones are alluded to. As an example, someone wishing to build on this very important work might be interested in a breed that was in the least seen category and having that referenced in Table 3 would provide context for them.
Or, if the additions of the less seen breeds would be too cumbersome, it would be nice to have the full list of purebred types seen in the supplementary section, which could then be referred to along with Table 3.
Answer: We agree with you, showing this data could be potentially valuable for future research. We added all the purebred types into the supplement table and have written in the results section the following: “Further affected breeds are listed in Table S4.”
Comment:
Regarding the designation of mixed breed versus purebred – in North America, we also affectionately call mixed breed dogs mutts. I realized after reading the author response that what I was actually unclear about is how dogs were assigned to this category by the authors for this study as being purebred or of mixed breed. The answer might be in this sentence on lines 134 to 135:
'The breed, sex and age of the included population were compared to data from the Swiss dog’s registration system (ANIS) and institutional records.'
Did the owners disclose that the dogs were of mixed race or purebred?
Answer: Yes, the owners disclosed this.
Comment:
Next, on page 6 I had asked: Only one dog was not admitted but triaged away (grade 1). Triaged right away? Is there a word missing?
- “Triage away” is a term used in veterinary medicine, when the emergency veterinarian assesses the patient as not sick enough for an admission to the hospital. In other words, the patient is deemed not to need immediate, intensive medical intervention and is instead given advice or treatment recommendations to follow outside the hospital setting. Here is a reference explaining this process: https://pubmed.ncbi.nlm.nih.gov/1586426/
Thank you for clarifying. I work closely with veterinarians here in North America and was unfamiliar with the term, so I recommend rewording to:
‘Only one dog was not admitted but rather assessed by the emergency veterinarian as not being sufficiently sick for admission (i.e., triaged away)’
This is a bit longer, but ensures all readers are clear on the circumstances for this dog.
Answer: Thank you, we have added this sentence.
Comment:
Finally, I greatly appreciate the caveat language added by the authors on lines 328-331. Would they consider creating a new paragraph, including the additional language they provided to me in their cover letter? Something like:
'Purebred dogs may have a more homogenous genetic pool than mixed dogs which may impact their immune response to insect stings. Due to the low prevalence of individual breeds we could not perform the statistical analysis for the individual breed. Showing that pure breed is associated with the disease points toward the possibility of a hereditary risk factor. In humans, hereditary alpha-tryptasemia is a risk factor for Hymenoptera venom allergy. Future research should consider analyzing specific breed characteristics in larger and more diverse populations to assess whether individual dog breeds show distinct patterns of HVA, thus clarifying if particular breeds have unique predispositions due to genetic or behavioral traits.'
Answer: We added this sentence and also a human reference to for the hereditary alpha tryptasemia.
